# A Prospective Validation Study of the Functional Bedside Aspiration Screen with Endoscopy: Is It Clinically Applicable in Acute Stroke?

**DOI:** 10.3390/jcm11237087

**Published:** 2022-11-29

**Authors:** Rebecca Kassubek, Beate Lindner-Pfleghar, Ioanna Eleni Virvidaki, Jan Kassubek, Katharina Althaus, Antonia Maria Weber, Georgios Dimakopoulos, Haralampos Milionis, Grigorios Nasios

**Affiliations:** 1Department of Neurology, University of Ulm, 89081 Ulm, Germany; 2Department of Speech and Language Therapy, University of Ioannina, 45500 Ioannina, Greece; 3Biostats, Epirus Science and Technology Park Campus, University of Ioannina, 45500 Ioannina, Greece; 4Department of Internal Medicine, School of Medicine, University of Ioannina, 45110 Ioannina, Greece; 5Department of Neurology, School of Medicine, University of Ioannina, 45110 Ioannina, Greece

**Keywords:** stroke, swallowing risk, aspiration, swallow screen, nutritional needs, FEES

## Abstract

The purpose of this study was to investigate the reliability of the novel Functional Bedside Aspiration Screen (FBAS) to predict aspiration risk in acute stroke and to guide initial therapy needs. We conducted a prospective validation study of the FBAS 10-point scale in 101 acute ischemic stroke patients. Outcome measures were compared with the Penetration Aspiration Scale (PAS) via the Flexible Endoscopic Swallowing Study. Correlations with the Functional Oral Intake Scale (FOIS) and the Therapy Requirement Scale (TRS) were analyzed. We observed a 65.8% sensitivity and 70.2% specificity (*p* = 0.004) for predicting penetration risk (for PAS score ≥ 3) and a 73% sensitivity and 62% specificity for predicting aspiration risk (PAS score ≥ 6). For patients with a modified ranking scale 0–2 (*n* = 44) on admission, the predictive measurements of the FBAS yielded sensitivity and specificity values of 66.7% and 88.6% (*p* = 0.011). A significant negative correlation was found with PAS measurements, whereas a positive correlation was observed regarding FOIS. Significantly lower FBAS scores were observed in patients with high requirements for therapeutic interventions and dietary modification. FBAS may be regarded as an alternative time-efficient clinical support tool in settings in which instrumentation is not directly accessible. Further studies including a larger cohort of acute stroke patients with more severe neurological deficits are necessary.

## 1. Introduction

Stroke is the most frequent acute neurological disease that leads to disruptions of the complex sensorimotor tasks involved in the act of deglutition, so that post-stroke dysphagia (PSD) may appear in at least 40% of patients within hours to days [1]. Affected patients have a four-fold increased risk of aspiration pneumonia and show significantly increased mortality and morbidity [2]. It is well-established that hospital settings that adhere to formal screening protocols can significantly lower their rates of stroke-associated pneumonia [3], and that conducting an aspiration screen is time-critical [4,5,6,7].

The utmost importance of dysphagia screening after acute stroke is to avoid adverse dysphagia-related pulmonary complications such as pneumonia and poor clinical outcomes that would impact patients’ health and social burden [2,8,9]. Still, it has to be considered that unnecessarily withholding oral alimentation or placing unwarranted nasogastric tube feeds can further increase patient dissatisfaction, caregiver burden, and health-related costs [10,11,12]. Improving the ability to diagnose aspiration risk early and accurately in acute stroke survivors through the implementation of standardized diagnostic algorithms may result in a more rapid and safe reintroduction of oral nutrition for patients who are not aspirating and could initiate appropriate rehabilitation techniques for those patients who are aspirating [12,13].

Water swallow tests, multi-consistency tests, and the Swallow Provocation Test [14] are methodologically validated in acute stroke. Most bedside screens rely on a 90 cc consecutive water challenge to rule out dysphagia or choking risk [15,16,17,18], but there is a need to account for the relative safety of a patient when asked to complete an interrupted drinking task, especially with an already compromised respiratory and immune system. Although the choice of the screening protocol frequently depends on patient characteristics and the availability of dysphagia specialists or further instrumental diagnostic procedures, a multi-consistency test may be preferred if swallowing efficiency is also to be assessed in addition to the safety of swallowing to guide initial diet recommendations within a reasonable time frame. A literature review highlights that stroke patients are at an increased likelihood of aspirating liquids compared to semi-solids [19]. As such, time-efficient, multi-consistency tests such as the as the Gugging Swallowing Screen (GUSS) [20] incorporate other consistencies aside from liquids in a graded, stepwise fashion. Similarly to the GUSS, the recently developed Functional Bedside Aspiration Screen (FBAS) [10] evaluates and necessitates patients’ management of saliva for suboral trials, as this clinical variable has been proven to be an independent and sensitive predictor of aspiration and aspiration pneumonia [21,22], although in our expertise, disturbed saliva management might be difficult to assess clinically. Additionally, the FBAS considers clinical information with regard to a patient’s functional language comprehension as well as the presence of dysarthria, with the latter proven to be a strong predictive factor for both dysphagia and aspiration risk [23,24,25].

Furthermore, both a critical and pragmatic outcome of bedside and instrument-based dysphagia assessment is estimating a patient’s relative nutritional risk after stroke and documenting his/her ability for functional oral intake with measures such as the Functional Oral Intake Scale (FOIS) [21]. In a recent attempt to replicate the validity of the GUSS in a larger cohort of acute stroke patients [26], Warnecke and colleagues found that diet recommendations were more conservative than those after FEES, and in particular, the GUSS seemed to overestimate the need for a ‘nil-per-os’ recommendation and nasogastric tube feeding. Additional patient parameters, including compliance, alertness, and ability to follow instructions, have to be taken into account during bedside screening [27], in addition to the PAS grade, in order to guide initial dietetic recommendations.

The objective of the current study was two-fold: We first aimed to examine the reliability of the FBAS to predict penetration and aspiration risk in acute stroke patients and its ability to constitute an alternative clinical support tool for frontline clinicians. Second, we intended to evaluate correlations between the FBAS scale and initial diet recommendations, supervision, and treatment requirements.

## 2. Materials and Methods

All consecutive stroke patients admitted at the Department of Neurology of the University Hospital of Ulm between 1 February 2021 and 6 July 2021 were evaluated. All patients were admitted to the emergency unit of the Department of Neurology, University of Ulm, Ulm, Germany, with a diagnosis of acute stroke. Within standardized operational procedures for stroke admission [28], all patients received a neurological examination (by a board-certified neurologist) and imaging immediately (within 30 min) after admission.

The study protocol was approved by the Ethics Committee of the University of Ulm (reference number 452/20). All recruited patients or their next-of-kin were informed about the study procedure and provided their written consent or assent where appropriate. Inclusion and exclusion criteria were consistent to our original validation study of the FBAS with the Yale Swallow Protocol [10].

Patients with hemorrhagic stroke were not included in this patient cohort. When compared to patients after an ischemic stroke, the dysphagia presented after a hemorrhagic stroke is frequently due to additional severe neurological impairments [29,30,31]. The time window in which swallowing status can be estimated after a hemorrhagic stroke is also highly variable due to frequent complications that may further delay bedside screening. Furthermore, patients with hemorrhagic stroke are more likely to fail a dysphagia screening—95% in the study of Joundi and colleagues [32,33]. There are no specific risk factors referred to dysphagia that will cause a failure in dysphagia screenings.

### 2.1. Functional Bedside Aspiration Screen (FBAS)

The FBAS was conducted within the first 24 h in 75.2% of our cohort and was performed according to the three steps followed in the original publication [10]. For the rest of our subjects, the latency of FBAS conduction ranged between 1–3 days. Step 1 included criteria for deferring oral trials, and step 2 marked patient-oriented clinical parameters that further served as negative or positive predictors for aspiration risk and were scored as either severely disturbed or adequate/“within normal limits” (WNL). The final step 3 of the protocol included gradually increased volumes of direct swallow trials. Failure criteria were inability to drink the entire amount, interrupted drinking, and a clear change in voice quality or coughing during or immediately after completing the task. Performance on the FBAS leads to a binary result (aspiration risk: high or low), although it is acknowledged that there may be different levels of severity of swallowing difficulty and subsequent different management needs. A score of >8 in our 10-point scale is a clinically significant cutoff point for reduced safety for oral feeding. Following the initial screen for aspiration risk, all patients were monitored for possible deterioration in their functional, general, and neurological status until their hospital discharge. In the present study, the FBAS preceded the FEES, which was carried out by two examiners blinded to the original FBAS score.

### 2.2. Other Swallowing Measures

In addition to the FBAS score, we used the commonly used Daniels predictors for internal validation. As originally proposed by Daniels and colleagues [34] and revisited by our group [10], the presence of two or more of 6 clinical features of the oropharyngeal mechanism predicts the presence of aspiration in acute stroke patients.

### 2.3. Functional Oral Intake Scale (FOIS)

The FOIS is a 7-point ordinal swallowing measure developed primarily in patients with stroke to document the functional eating ability of food and liquid by mouth. Levels 1 through 3 indicate tube dependency, and levels 4 through 7 relate to total oral intake. The FOIS needs no specialized training for clinicians familiar with the management of stroke patients. This functional standardized scale has been shown to have adequate reliability, validity, and responsiveness to detect changes over time [19].

### 2.4. Flexible Endoscopic Evaluation of Swallowing (FEES)

All patients underwent FEES, which was used as a reference standard to identify aspiration risk and was performed according to standardized procedures [10] using a 2.9 mm RS1 CCD Video Rhino Laryngoscope by Orlvision^®^ (Lahnau, Germany) recorded with the software rp Szene^®^ (Rehder/Partner GmbH, Hamburg, Germany). It was performed and evaluated by a trained and certified speech and language therapist and supervised by a specialized senior clinician (board-certified in neurology). The investigation followed a stepwise standardized in-house protocol as reported previously [35], including patients with nasogastric tubes. The following consistencies were evaluated: porridge (apple puree), fluid (water), nectar-like (banana nectar), soft (bread without crust), solid-mixed (apple), and a placebo–pill (together with water, nectar, or puree). Bolus volume of puree and liquids was gradually increased. In our study, we used stepwise increasing volumes, i.e., our protocol included one teaspoon and one tablespoon volume of puree. Water was administered in volumes of teaspoon and tablespoon, one single sip from a glass, followed by sequential/consecutive swallows from a glass. In the case of abnormal findings such as pronounced leaking, penetration, or aspiration, nectar was given in the same way. In addition, a bite of bread, a bite of apple, and a placebo pill accompanied with puree or liquid were part of the protocol. Termination criteria of each consistency and volume size was evidenced by the aforementioned dysphagia symptoms, which could not be compensated by instruction of the investigator.

Materials were dyed with methylene blue to highlight penetrations or aspirations. If any risk for aspiration was detected, the subsequent, larger bolus volumes were not tested. FEES was performed on the same day as the initial FBAS screening in 75.2% of our patient sample, the following day in 14.9%, and with a delay of more than one day in 9.9% of our cohort. A speech-language pathologist, blinded to protocol results, reviewed the FEES to determine aspiration status in a binary (yes/no) manner. For the semiquantitative evaluation of penetration and aspiration, the penetration aspiration scale (PAS) [36] was used in its German version [36,37]. Furthermore, significant spillage (posterior leaking into the laryngeal vestibule) and significant residue in the lower pharynx (valleculae, pyriform sinuses, postcricoid region) were documented [38].

### 2.5. Therapy Requirement Scale

As an expansion to the commonly used scales for the assessment of dysphagia severity, mainly based on PAS, we conducted a novel Therapy Requirement Scale (TRS). Based on the results of the FEES and other clinical findings such as compliance, alertness, apraxia, and neglect, this new four-grade rating scale for therapeutical requirement assesses the need and extent of therapeutic interventions, counseling/education, and supervision, i.e., information that cannot be deduced from the presence or absence of penetration and aspiration alone. TRS, as shown in Table 1, can be explained as follows: TRS 0 indicates normal swallowing function without any further requirements; TRS grade 1 means at least educational guidance with regards to the necessary compensatory strategies or diet changes and occasional dysphagia therapy as needed; TRS grade 2 is described as the necessity of intensive dysphagia therapy to advise compensatory strategies or diet modification as well as functional training; TRS grade 3 suggests profound diet recommendation in the sense of full or partial tube feeding, frequent intensive dysphagia therapy, education, and guidance.

### 2.6. Statistical Analysis

The FBAS scores as well as all the scale variables were expressed as means and standard deviations. All categorical variables were expressed as counts and percentages. ROC curves were used to estimate the sensitivity and specificity of the FBAS compared to the FEES outcomes. Subgroup estimations were carried out to examine changes in these estimates based on mRS. The Pearson correlation index was used to assess the relationship between FBAS and FOIS after data skewness analysis. Differences in the FBAS values were also examined based on the TRS value using analysis of variance and the Bonferroni correction for multiple comparisons. Chi square tests were applied to examine independence between FBAS risk and PAS binary assessment as well as the independence between Daniels outcome and PAS binary assessment. Statistical analysis was conducted using the SPSS v 26.0. Significance was set at 0.05 in all analyses.

## 3. Results

The clinical characteristics of the study population are summarized in Table 2, together with correlations with PAS values. Of the total of 110 patients, 9 were excluded from our final cohort. In detail, four patients were removed from final data analysis due to dysphagia prior to the ischemic stroke, and two patients presented with a significant deterioration in their clinical state and could not undergo endoscopic swallowing evaluation but required treatment in another department. Finally, initial stroke diagnosis was not confirmed in two patients, and one patient withdrew informed consent.

A total of 101 prospectively enrolled patients with ischemic stroke (53 females and 48 males, mean age 71.0 years, sd = 12.25) were included in the final statistical analysis. A two-by-two contingency table was used to compare the results of the FBAS with the results of aspiration or non-aspiration on the FEES studies (Table 3). All patients underwent the FBAS protocol without complications. The mean FBAS score was 8.29.

The ROC curve analysis depicted a statistically significant discriminant ability of the FBAS protocol with regards to FEES (Figure 1). Specifically, we found a 65.8% sensitivity and a 70.2% specificity (area under the curve, AUC = 0.67, 95% CI 0.56–0.781; *p* = 0.004) for predicting penetration risk (in the case of PAS score ≥ 3) and a 73% sensitivity and 62% specificity for predicting aspiration risk (in the case of PAS score ≥ 6). With respect to a subcategory of the patients with mRS at entry 0, 1, 2 (*n* = 44), the predictive measurements of the FBAS yielded sensitivity and specificity values of 66.7% and 88.6%, respectively (AUC = 0.778, 95% CI 0.58–0.976; *p* = 0.011).

In this patient cohort with mostly mild-to-moderate neurological deficits, most patients showed pronounced leaking, penetration, or aspiration after more than teaspoon volumes (12/15). Three patients showed signs of aspiration after sequential consecutive swallows from a glass, five after a single sip from a glass, and three more after a tablespoon volume. Those patients would not have been identified in the previously published studies, since they did not show aspiration in FEES with teaspoon volumes. Only one patient with aspiration of a teaspoon of nectar had a higher FBAS score of seven; one other patient with aspiration of teaspoon volumes in the FEES had a FBAS score of four, and another patient who could not be tested for oral intake due to obviously too high a risk of aspiration had an FBAS score of two.

The FBAS showed a positive and moderately strong correlation with the FOIS as depicted in Figure 2. Additionally, differences in the FBAS scale were observed depending on the requirement for swallowing therapy according to the TRS. Specifically, significantly lower FBAS scores were observed in TRS2 and TRS3 patients as compared to TRS0 or TRS1 patients. Differences between TRS0 and TRS1 patients as well as between TR2 and TRS3 patients were not statistically significant (Figure 3).

The significant correlation between TRS and PAS as well as FOIS is shown in Table 4.

## 4. Discussion

The purpose of this prospective study was to investigate an agreement for aspiration risk in the same individuals with acute stroke between the FBAS and the FEES gold standard and to evaluate correlations between outcome on the FBAS and initial nutrition or dysphagia therapy needs. When using the reference standard of FEES, the discriminant ability of the FBAS to document penetration and aspiration risk as defined by the PAS scale was significant. On the other hand, the FBAS had a lower sensitivity to detect aspiration compared to other clinical assessments. Here, the very detailed FEES protocol (as described in the methods section) was a relevant confounder, with examination of not only teaspoon volumes, as has been commonly done in previous studies [20,26,39,40]. In detail, 12 out of 15 patients with a PAS score ≥6 would not have been identified in the previously published studies, since they showed aspiration in FEES only after volumes higher than teaspoon.

Furthermore, this limited sensitivity has to be seen in the context that a bedside screening for aspiration is needed for preventing stroke-associated pneumonia. It is of note that a higher stroke-associated pneumonia rate is reported not only in stroke patients who failed a high-sensitive bedside screening compared to those who passed the screening [41], but more importantly, in stroke patients who passed a low-sensitive screening for dysphagia compared to those who passed a high-sensitive screening [42]. In our cohort, we did not have any case of stroke-associated pneumonia, possibly attributed to the fact that FEES was performed in each and every patient.

The FBAS score correlated negatively with the FOIS and showed a linear trend to the need for special dysphagia guidance and restrictions of oral intake (according to TRS). Therefore, FBAS might serve as a first screening instrument to identify patients with high risk for dysphagia with aspiration who therefore have a high demand for further intensive, close-meshed surveillance and logopedic therapy. Further studies including a larger cohort of acute stroke patients are necessary to verify the promising tendency of linear correlation between FBAS and FOIS as well as the demand for therapeutic interventions. The introduction of the novel Therapy Requirement Scale (TRS) in this context addresses some major clinical aspects in dysphagia management after acute stroke, i.e., prevention of swallowing-related complications and initiation of rehabilitation as soon as possible. The four-point scale of the TRS is able to contribute to the assessment of these clinical needs in a systematic quantitative approach that is simple to apply in a clinical setting. The methodology and frequency/intensity of swallowing therapy can be assessed from the clinical presentation and the technical investigations. The TRS can help to improve the (interdisciplinary) communication about the organizational management of therapy measures for dysphagia, including allocation of resources. The further validation of the TRS is a future task that remains to be addressed in a multi-site study.

One strength of our study was the specifically differentiated FEES protocol, better representing normal mealtime and drinking. Taken together, this FEES protocol may suggest a lower sensitivity of FBAS than published in other clinical assessments before, for example, GUSS [24], but remarkably, specificity was higher despite this detailed protocol. Other protocols, for example, GUSS, tend to overestimate the need for nasogastric tube feeding. Screenings with a binary concept such as the Yale Swallow protocol tend to recommend NPO in case of any failure [24] and thus lead to overestimating the demand of a nasogastric tube, whereas FBAS may lead to less conservative diet recommendations and less unneeded nasogastric tube feeding. Is it of note that such a result from FBAS should be used with caution in clinical practice, given that the scope of dysphagia and aspiration monitoring is mostly avoiding respiratory complications such as pneumonia [2,8,9].

The current study had a similar design as a recent study investigating the Sapienza Global Bedside Evaluation of Swallowing after Stroke (GLOBE-3S), which combines the Toronto Bedside Swallowing Screening Test with oxygen desaturation and laryngeal elevation measurement during swallowing in 50 stroke patients [43]. The authors report that GLOBE-3S could represent a highly sensitive instrument to avoid the misdiagnosis of silent aspirators by including the measurement of laryngeal elevation and monitoring of oxygen desaturation. The conclusions by Toscano and colleagues are in agreement with our study, given that the semiquantitative PAS values constitute only a partial aspect with the detection threshold as the reference. Additional symptoms such as leaking or pharyngeal retention as well as individual factors such as orofacial functional impairment or (pre-existent or stroke-associated) cognitive deficits are important for determining the therapy requirements.

Besides an adequate swallowing reflex, many recent studies highlight that evaluating a patient’s defensive mechanism against airway invasion is important in identifying patients at higher risk for silent aspiration and ultimately in preventing aspiration pneumonia [44,45,46,47]. A patient’s possible impairment of the cough reflex and other abnormal protective responses (such as throat clearing) in swallowing trials are incorporated measures in the FBAS protocol. On a further note, an ideal protocol might not only refer to aspiration but, likewise, also to considerable dysphagia symptoms with the need for therapy, surveillance, and special diet recommendations. As one further example, an obvious impairment of saliva management might be one important indicator for disturbed swallowing function. FBAS might be on step towards this.

Our study was not without limitations. In general, we had an underrepresentation of major strokes that would suggest a high aspiration risk, which very often will be confirmed in FEES, then leading to higher sensitivity rates, as shown in previous studies. Although we did not find a highly sensitive prediction of aspiration by FBAS, our data showed a tendency in that respect, which may be confirmed by further studies with larger cohorts of both minor and major strokes. The lower the FBAS score, the higher the PAS scores were. For the Gugging Swallowing Screen revisited [26], Warnecke et al. assessed a sensitivity of 100% in patients with NIHSS of >15, whereas the sensitivity for patients with a NIHSS < 5 was only 71.4%. It seems safe to suppose that patients with severe strokes will mostly not even reach FBAS step 3 with swallowing tests and thus achieve a low FBAS score, indicating a high aspiration risk. This group is lacking in our study but is expected to be correctly identified as patients with a high risk for dysphagia.

A further limitation of our study design is based upon the fact that dysphagia in patients with stroke is not only strongly associated with stroke severity but may even be transient in patients with low NIHSS, as demonstrated by Toscano and colleagues [48]. For these patients with low NIHSS, frequent dysphagia screenings and re-screenings are mandatory, so that it has to be considered that a higher latency between the first clinical assessment and FEES can confound the results due to improving dysphagia. As most of our patients had low NIHSS, but 10 percent of our study population had more than 24 h delay between FBAS and FEES, this latency may be regarded as a confounder to the results in this subgroup.

Nevertheless, clinical assessments are always limited tools to safely identify all patients with aspiration risk after ischemic stroke [49]. There is a strong recommendation to establish instrumental diagnostic techniques, especially FEES, if possible. Considering that aspiration pneumonia is a preventable complication that continues to occur in many patients after stroke, early estimates of swallowing-related aspiration screenings as well as full clinical bedside examinations should be used to identify patients at a high risk for dysphagia and to perform further investigation immediately on a low threshold [13].

Finally, as previously pointed out by Khorsan and colleagues [50], external validity is often neglected in methodological considerations in healthcare research. Another strength of our two-site study is that the outcome analyses were carried out by researchers in different geographical healthcare settings who were blinded to the data collectors’ assessment procedures, thus ensuring reduced bias in our study results and conclusions. However, the relatively small size of our patient sample could not characterize subcategories of stroke severities. Therefore, the discriminant ability of FBAS is yet to be further explored with larger, and thus more representative, samples of patients.

## 5. Conclusions

The clinical validity, reliability, and usefulness of the FBAS to determine aspiration risk has been confirmed in this study. This protocol probably leads to less conservative diet recommendations and therefore allows for timely oral nutrition, hydration, and medications, with the need for frequent re-evaluation of swallowing function. FBAS may be regarded as an alternative time-efficient clinical support tool in settings in which instrumentation is not directly accessible. Nevertheless, the necessity for instrumental diagnostic procedures such as the FEES in acute stroke-associated dysphagia has again been confirmed. Further studies including a larger cohort of acute stroke patients with more severe neurological deficits are necessary to verify the promising tendency of linear correlation between FBAS and FOIS, as well as the demand for therapeutic interventions.

## Figures and Tables

**Figure 1 jcm-11-07087-f001:**
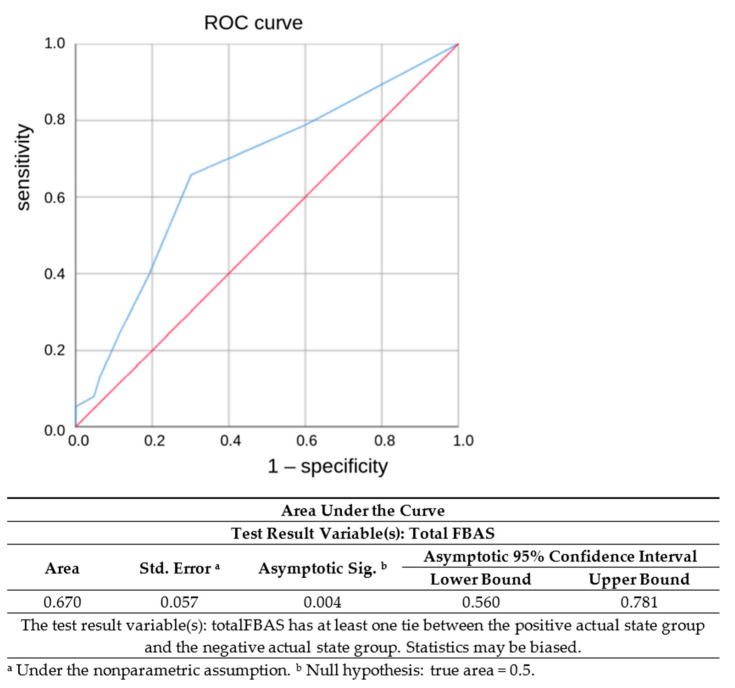
Receiver operating characteristic (ROC) curve for the ability of the Functional Bedside Screen (FBAS) to predict aspiration risk in ischemic stroke.

**Figure 2 jcm-11-07087-f002:**
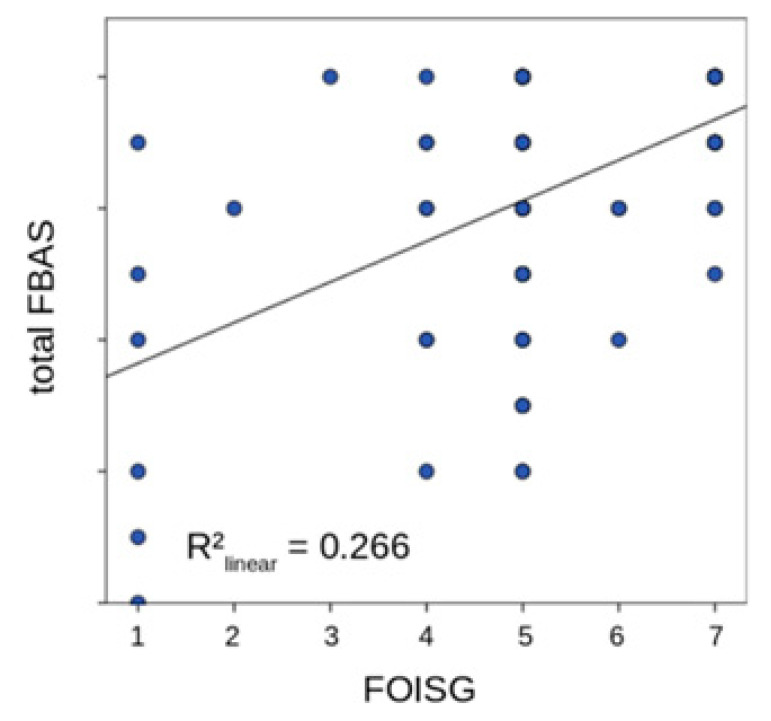
Correlation of the Functional Bedside Aspiration Screen (FBAS) to Functional Oral Intake Scale (FOIS).

**Figure 3 jcm-11-07087-f003:**
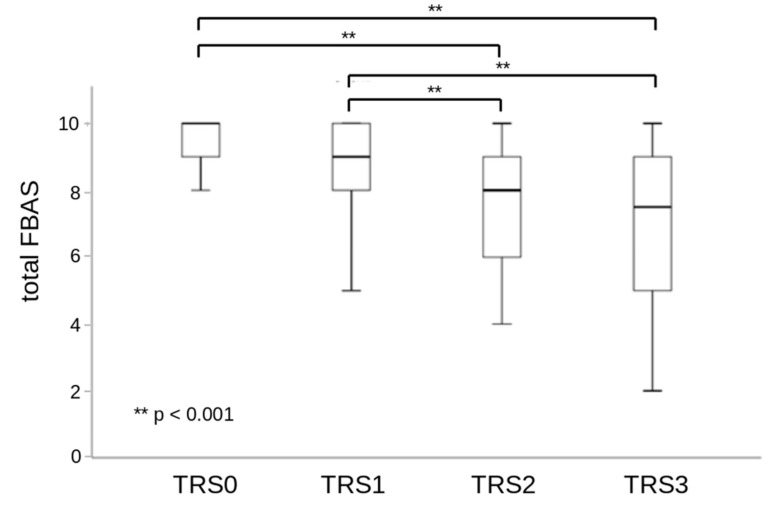
Comparative boxplots of the Functional Bedside Aspiration Screen (FBAS) with the need for specialized swallowing therapy (TRS). TRS0 = no dysphagia, no counseling or diet modification required; TRS1 = education and guidance with regards to the necessary compensatory strategies or diet changes, as needed occasional dysphagia therapy; TRS2 = compensatory strategies or diet changes required, intensive dysphagia therapy, education and guidance; TRS3 = profound diet changes required, full or partial tube feeding, intensive dysphagia therapy, education and guidance.

**Table 1 jcm-11-07087-t001:** Therapy Requirement Scale.

Grade	Therapy Requirement Scale
TRS 0	no dysphagia, no counseling or diet modification required
TRS 1	education and guidance with regard to the necessary compensatory strategies or diet changes, as-needed occasional dysphagia therapy
TRS 2	compensatory strategies or diet changes required, intensive dysphagia therapy, education and guidance
TRS 3	profound diet changes required, full or partial tube feeding, intensive dysphagia therapy, education and guidance

**Table 2 jcm-11-07087-t002:** Patients’ clinical characteristics.

		Total *n* = 101	PAS 1–2 *n* = 63	PAS 3–5 *n* = 23	PAS 6–8 *n* = 15	*p*-Value
Sex	Female	53	32	16	5	0.083
	Male	48	31	7	10	
NIHSS at admission	Mild (0–4)	66	46	10	10	0.018
	Moderate (5–15)	32	15	13	4	
	Moderate to severe (16–20)	1	0	0	1	
	Severe (21–42)	1	1	0	0	
mRS atentry	0—No symptoms	5	4	1	0	0.007
	1—No significant disability	28	22	3	3	
	2—Slight disability	11	9	1	1	
	3—Moderate disability	24	14	5	5	
	4—Moderate to severe disability	13	5	5	3	
	5—Severe disability	20	9	8	3	
Hemisphere	Right	41	26	10	5	0.201
	Left	39	26	5	8	
	Bilateral	21	10	8	2	
Arterial supply area of stroke	ACA	0	0	0	0	0.252
	MCA	51	33	12	5	
	PCA	7	4	0	2	
	VBA	21	13	3	5	
	Multiple vascular territories	22	11	8	2	
Screening	FBAS positive (score 0–8)	44	19	14	11	0.002
	FBAS negative (score 9–10)	57	44	9	4	
	pneumonia	0	0	0	0	

Abbreviations: NIHSS, National Institute of Health Stroke Scale; mRS, modified Rankin scale; ACA, anterior cerebral artery; MCA, middle cerebral artery; PCA, posterior cerebral artery; VBA, vertebrobasilar arteries.

**Table 3 jcm-11-07087-t003:** Total number of false positives, false negatives, true negatives, and true positives for the Functional Bedside Aspiration Screen (FBAS; 2 × 2 contingency table).

Binary PAS × FBASrisk				
		FBASrisk		Total
		Low	High	
Binary PAS	Low	TN = 44	FP = 19	63
	High	FN = 13	TP = 25	38
Total		57	44	101

Note: True positive = failed FBAS and aspiration on fiberoptic endoscopic evaluation of swallowing; false positive = failed FBAS and no aspiration observed on fiberoptic endoscopic evaluation of swallowing; false negative = pass for FBAS and aspiration observed on fiberoptic endoscopic evaluation of swallowing; true negative = pass for FBAS and no aspiration on fiberoptic evaluation of swallowing.

**Table 4 jcm-11-07087-t004:** Correlations between TRS and PAS and FOIS.

Correlations		
		TRS
PAS	Pearson Correlation	0.687
	Sig. (2-tailed)	0.000
	N	101
FOISG	Pearson Correlation	−0.849
	Sig. (2-tailed)	0.000
	N	101

## Data Availability

The data that support the findings of this study are available from the corresponding author upon reasonable request.

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
