# Peer review of "A Prospective Validation Study of the Functional Bedside Aspiration Screen with Endoscopy: Is It Clinically Applicable in Acute Stroke?"

_jcm, 2022, doi:10.3390/jcm11237087_

Round 1
Reviewer 1 Report
The paper by Kassubeck et al deals with the implementation of a bed side test for aspiration and dysphagia in acute stroke patients. The subject is quite interesting being the development of bed side tests an open and useful field. I have some points that remain unclear to me.
Methodologically, I find results presentation confusing. Many analyses are presented but not all are displayed. And instead sub analysis which are not presented in methods are given in the results (as the re-run of ROC curves according to PAS scores). In the text, I do not see links to supplements to get to other results. Besides this strange choice (which should be fixed) I do not understand which is the added value of increasing the number of analysis and tests? One example is why authors used Pearson’s correlation between scales used, to which purpose. As well the analysis about TRS
In the table 2 chi-square test is present. Besides that no significance level is provided for this table (btw the contingency table is significant), I do not see why authors used this test and not an inter-rater reliability measure.
In the theoretical frame, I do not understand authors’ premise (which is also reprised in the discussion) to find a scale that will not overestimate the need for nasogastric tube per se. The scope of dysphagia and aspiration monitoring is mostly avoiding complication that would impact on patients’ and social burden. In this sense, I recommend to see this articles PMID: 31226922, PMID: 30199856, PMID: 34746431.
Moreover, one limitation that is not stated is that data show that dysphagia and aspiration risk is transient (about a week in low NIHS patients as those recruited in this study) so an adequate test with quick re-evaluations could be preferable to a moderate precise test. In this view, also a few days delay between bedside and FESS could skew results. I think that this point should be explained (you can refer to PMID: 26492033 for reference).
When compared with results from other studies, authors did not include the paper PMID: 30414300 which possibly is the most recent paper on this subject and also has a very similar design to the present study so it is very strange to understand why authors didn’t include this paper.
In the discussion, lines from 245 to 249 refer to data not presented in the paper so should be removed on authors should include data they are referring to.
Author Response
To the Editorial Team of Journal of Clinical Medicine, MDPI
Dear Ms. Mance Shi,
We wish to express our appreciation to the Editorial Board and the Reviewers for the careful consideration of our manuscript No. jcm-2022508, entitled ‘A prospective validation study of the functional bedside aspiration screen with endoscopy: Is it clinically applicable in acute stroke?’ and for inviting us to resubmit a version incorporating the revisions suggested by the reviewers. Their valuable feedback helped us to introduce significant improvements and enrich our reference list. Please note the detailed point-by-point response to the reviewers’ comments below.
Enclosed, please find the revised version of the manuscript where all changes have been highlighted in yellow. We thank you again for your time and feedback and we look forward to your final positive response to our revised manuscript.
Sincerely yours,
Dr. Rebecca Kassubek
Corresponding Author
Response to reviewers’ comments
We wish to express our appreciation to the Reviewers for the careful consideration of our manuscript and their insightful comments, which have helped us significantly to improve our manuscript. Enclosed please find the revised version of the manuscript according to the suggestions. Point-by-point responses to the reviewers’ comments are listed below. For readability purposes, we have colored blue the reviewers’ comments.
Reviewer I:
The paper by Kassubeck et al deals with the implementation of a bed side test for aspiration and dysphagia in acute stroke patients. The subject is quite interesting being the development of bed side tests an open and useful field. I have some points that remain unclear to me.
Authors’ response: Thank you for your helpful comments and for taking the time to point out options to improve our manuscript.
Comment 1: Methodologically, I find results presentation confusing. Many analyses are presented but not all are displayed. And instead sub analysis which are not presented in methods are given in the results (as the re-run of ROC curves according to PAS scores). In the text, I do not see links to supplements to get to other results. Besides this strange choice (which should be fixed) I do not understand which is the added value of increasing the number of analysis and tests?
- One example is why authors used Pearson’s correlation between scales used, to which purpose
- As well the analysis about TRS
Authors’ response: Thank you for calling our attention to the correlation estimates of the FBAS presented in our manuscript. As suggested, the dot plot of the FBAS with PAS in Figure 2 has been removed since the ROC curve illustrates the discriminant ability of the FBAS against FEES (with regards to PAS). However, we do believe that estimating a patient’s nutritional risk after stroke and documenting his/her ability for functional oral intake with FOIS (Crary MA et al., 2005) is both a pragmatic and an important outcome of screening since patients must safely receive oral meds and return to an oral diet as soon as possible. This may not only minimize time on artificial feeding methods but also help them receive the most appropriate nutritional intake to assist in their rehabilitation post stroke. In this regard, the correlations of FOIS with FBAS and the need for dysphagia counseling or specialized swallowing therapy as depicted by the TRS scale are important outcomes of swallowing screening after stroke. Please note that we rephrased and reordered the sentence in lines 85-86 to the beginning of the paragraph. (Please also refer to our answer regarding TRS scale to Reviewer 2 Comment 7.)
“Furthermore, a critical, as well as pragmatic, outcome of bedside and instrument-based dysphagia assessment is estimating a patient’s relative nutritional risk after stroke and documenting of his/her ability for functional oral intake with measures such as the Functional Oral Intake Scale (FOIS) (Crary MA et al., 2005).”
Comment 2: In the table 2 chi-square test is present. Besides that no significance level is provided for this table (btw the contingency table is significant), I do not see why authors used this test and not an inter-rater reliability measure.
Authors’ response: Thank you for your comment and the opportunity to further clarify the need for this table. Please note that we did not calculate a chi-square, but instead, we present the estimates used in the ROC analysis. We did expect that we would not have a high or an absolute agreement between the FBAS and the PAS scale. The FBAS is meant to be a functional bedside screening tool leading to a binary outcome (low or high risk) in the acute phase of stroke management and not to represent a substitute for the Penetration Aspiration Scale used in conjunction with instrumental swallowing assessments (videofluoroscopy/endoscopy) to characterize the depth and response to airway invasion and to guide dysphagia treatment based on patients’ observed swallowing physiology. Under this perspective, we believe that an agreement coefficient does not serve the tables’ purpose, which is actually informative regarding the ROC analysis that follows.
Comment 3: In the theoretical frame, I do not understand authors’ premise (which is also reprised in the discussion) to find a scale that will not overestimate the need for nasogastric tube per se. The scope of dysphagia and aspiration monitoring is mostly avoiding complication that would impact on patients’ and social burden. In this sense, I recommend to see this articles PMID: 31226922, PMID: 30199856, PMID: 34746431.
Authors’ response: We thank the reviewer for pointing out this important item. Accordingly, we have revised the text in the introduction and the discussion.
The revised introduction reads:
“The utmost importance of dysphagia screening after acute stroke is to avoid adverse dysphagia-related pulmonary complications like pneumonia and poor clinical outcome that would impact on patients’ health and social burden (Eltringham et al., 2018; Ouyang et al., 2020; Dziewas et al., 2021). Still, it has to be considered that unnecessarily withholding oral alimentation or placing unwarranted nasogastric tube feeds can further increase patient dissatisfaction, caregiver burden, and health-related costs.”
The discussion reads:
“Screenings with a binary concept like the Yale Swallow protocol tend to recommend NPO in case of any failure [24] and thus lead to overestimate the demand of nasogastric tube, whereas FBAS may lead to less conservative diet recommendation and less unneeded nasogastric tube feeding. Of note, such a result from FBAS has to be used with caution in clinical practice, given that the scope of dysphagia and aspiration monitoring is mostly avoiding respiratory complications like pneumonia (Eltringham et al., 2018; Ouyang et al., 2020; Dziewas et al., 2021).”
This aspect was also toned down in the conclusion, reading:
“This protocol probably leads to less conservative diet recommendation and therefore allows for timely oral nutrition, hydration, and medications, with the need for frequent re-evaluation of swallowing function.”
We have included the three references as proposed by the reviewer.
Comment 4: Moreover, one limitation that is not stated is that data show that dysphagia and aspiration risk is transient (about a week in low NIHS patients as those recruited in this study) so an adequate test with quick re-evaluations could be preferable to a moderate precise test. In this view, also a few days delay between bedside and FESS could skew results. I think that this point should be explained (you can refer to PMID: 26492033 for reference).
Authors’ response: We fully agree with the reviewer. In our clinical routine, patients with ischemic or hemorrhagic stroke should be bedside screened for dysphagia within 24 hours. FEES should be conducted at the same day if indicated. If dysphagia has been diagnosed by FEES, patients will be rescreened either clinically and if indicated by FEES within the following week. Ninety percent of our patients had FBAS and FEES within 24 hours, unfortunately ten percent had not. We revised the limitations paragraph accordingly, including the item of transient dysphagia as raised by the reviewer:
“… A further limitation of our study design is based upon the fact that dysphagia in patients with stroke is not only strongly associated with stroke severity but may be even transient in patients with low NIHSS, as demonstrated by Toscano and colleagues (Toscano et al., 2015). For these patients with low NIHSS, frequent dysphagia screenings and re-screenings are mandatory, so that it has to considered that a higher latency between first clinical assessment and FEES can confound the results due to improving dysphagia. As most of our patients had low NIHSS, but 10 percent of our study population had more than 24 hours delay between FBAS and FEES, this latency may be regarded as a confounder to the results in this subgroup.”
The new reference was included:Toscano, M.; Cecconi, E.; Capiluppi, E.; Vigano, A.; Bertora, P.; Campiglio, L.; Mariani, C.; Petolicchio, B.; Sasso D’Elia, T.; Verzina, A.; Vincenzi, E.; Fiorelli, M.; Cislaghi, G.; Di Piero, V.; Neuroanatomical, Clinical and Cognitive Correlates of Post-Stroke Dysphagia. Eur Neurol. 2015, 74 (3-4), 171-7.
Comment 5: When compared with results from other studies, authors did not include the paper PMID: 30414300 which possibly is the most recent paper on this subject and also has a very similar design to the present study so it is very strange to understand why authors didn’t include this paper.
Authors’ response: We thank the reviewer for calling our attention to this helpful reference. This study is in agreement with our data, given that we also consider additional symptoms like laryngeal penetration (PAS 3-5), leaking/premature spillage and pharyngeal residuals as important dysphagia symptoms besides confirmed aspiration (PAS ≥6), as now mentioned in the manuscript.
The discussion reads: “The current study had a similar design as a recent study the Sapienza Global Bedside Evaluation of Swallowing after Stroke (GLOBE-3S) which combines the Toronto Bedside Swallowing Screening Test with oxygen desaturation and laryngeal elevation measurement during swallowing in 50 stroke patients (Toscano et al., 2019). The authors report that GLOBE-3S could represent a highly sensitive instrument to avoid the misdiagnosis of silent aspirators by including the measurement of laryngeal elevation and monitoring of oxygen desaturation. These conclusions are in agreement with our study, given that the semiquantitative PAS values constitute only a partial aspect with the detection threshold as the reference. Additional symptoms like leaking or pharyngeal retention as well as individual factors like orofacial functional impairment or (pre-existent or stroke-associated) cognitive deficits are important for evaluating the therapy requirements.”
Comment 6: In the discussion, lines from 245 to 249 refer to data not presented in the paper so should be removed on authors should include data they are referring to.
Authors’ response: We agree with the reviewer that this paragraph was misplaced in the original version of the manuscript and have accordingly moved this text which contains results to the Results section. In addition, the paragraph in ll. 236-244 of the original submission which explained this text has been moved to the methods.
As a consequence, as per reviewer’s suggestion, only the interpretation of these results remained in the discussion section, reading:
“Here, the very detailed FEES protocol (as described in the methods section) was a relevant confounder, with examination of not only teaspoon volumes as it has been commonly done in previous studies. In detail, 12 out of 15 patients with PAS score≥6 would not have been identified in the previously published studies since they showed aspiration in FEES only after volumes higher than teaspoon.”

Reviewer 2 Report
The prospective validation study by Kassubek and colleagues aims to investigate the reliability of the FBAS 10‐point scale to predict aspiration risk in acute stroke patients in 101 acute ischemic strokes. They observed a 73% sensitivity and 62% specificity 22 for predicting aspiration risk (PAS score ≥6).
Strengths: This study adopted a differentiated FEES protocol which better represents normal mealtime and drinking. In fact, in this study, the sensitivity of the FBAS was lower than published in other clinical assessments before. It is also worth noting that all patients underwent the FEES as the gold standard.
Thus, thanks to the study design, this prospective validation study can paint a very realistic picture of dysphagia and aspiration in acute stroke.
I also appreciated the focus on aspiration, because many bedside screenings are focused on dysphagia without considering aspiration and vice-versa. This is misleading, at best, because dysphagia may occur without aspiration and vice versa.
Thus, I have just some minor points.
In my opinion, the paper would be considerably improved by reviewing the following minor issues:
Weakness:
1) I am not sure at all that the observed sensitivity (73%) allows the FBAS to be a “valid first screening instrument to identify patients with high risk for dysphagia” as stated in the text (lines 257-8), even with high specificity.
By definition, a valid swallowing screening in the acute phase of stroke has to be sensitive enough to accurately identify stroke patients at risk of swallowing impairment. A high sensitivity, more than a high specificity, is required because of the increased morbidity and mortality associated with aspiration, which necessitates low false-negative results.
In this study, among 57 patients at low FBAS risk, 13 resulted as false-negative against FEES. This implies a negative predictive value of 77,2%, meaning that almost 1/4 of patients with aspiration will be tested as negative with the FBAS.
- Please, review this statement and report the already validated bedside screenings that have been designed to detect aspiration and achieved a higher sensitivity than FBAS when tested against the FEES (please see Martino R et al. Stroke. 2009;40(2):555-61 and Toscano M et al. Eur J Neurol. 2019 Apr;26(4):596-602).
2) I agree with the Author when claiming that one possible explanation for the FBAS's lower sensitivity to detect aspiration is that previous studies probably missed patients who did not show aspiration in FEES with teaspoon volumes.
Anyway, in my opinion, the main reason for the lower sensitivity observed is that the FBAS probably missed the silent aspirators, and likewise most of the bedside dysphagia screenings. Although overt dysphagia is a common feature in stroke patients, those may also suffer from silent aspiration (please see Daniels SK et al. Stroke. 2012;43(3):892-7). This has been linked to reduced laryngopharyngeal sensation, impaired cough reflex, low dopamine or substance P levels, and central/local weakness/incoordination of the pharyngeal musculature (see Toscano M et al. Eur J Neurol. 2019 Apr;26(4):596-602).
- Please, comment on silent aspiration in the discussion as a risk factor (mostly neglected) for PSD and aspiration pneumonia, as well as a possible explanation for the low sensitivity achieved by the FBAS.
- In this regard, since the FEES can detect silent aspiration, it could be of great interest to show data about that or at least, comment on it in the discussion.
3) In my opinion, when performing a validation study of a bedside screening for aspiration, it is mandatory taking into account the capability of this screening in preventing stroke-associated pneumonia (SAP).
The main reason is that patients with dysphagia may have an 11-fold higher risk of developing pneumonia than non-dysphagic patients, depending on several factors such as stroke severity, the presence of aspiration and silent aspiration, and most of all the kind of bedside screening performed in the acute stage.
In the specific, concerning the screening method, a higher SAP rate is reported: A) in stroke patients who failed a high-sensitive bedside screening compared to those who passed the screening (see Suntrup-Krueger S et al. Cereb. Dis. 2018;45(3-4):101-8). B) but more importantly, in stroke patients who passed a low-sensitive screening for dysphagia compared to those who passed a high-sensitive screening which can also detect both aspiration and silent aspiration. (see Jannini TB et al. Neurol Sci. 2022 Feb;43(2):1167-1176.).
- Please comment on the importance of BSE for preventing SAP in the introduction and discussion sections.
- Please, report the SAP rate in the different groups, otherwise add the lack of this data as a study limitation and highlight that SAPs may arise due, beyond to the aspiration and the kind of BSE, also to stroke-related factors, like immunodepression coupled with tube feeding-related oropharyngeal colonization of pathogenic organisms, especially in the ICUs, where SAP incidence is up to 38% (see Westendorp WF et al. BMC Neurol. 2011;11:110 - Hannawi Y et al. Cerebrovasc Dis 2013;35(5):430-43 - Kishore AK et al. Eur Stroke J. 2019;4(4):318-28).
4) Why did the Authors exclude hemorrhagic stroke, thus losing possible relevant data? In this regard, it is reported an increased frequency of dysphagia in hemorrhagic stroke patients compared to ischemic ones (Paciaroni M, et al.: Dysphagia following stroke. Eur Neurol 2004;51:162-167). Moreover, hemorrhagic stroke could be also an independent predictor of both post-stroke dysphagia and a persistent pattern of dysphagia at 14 days. Please, comment on this.
5) It is reported that FEES was performed with a delay of more than one day in 9.9% of patients (line 136). Since about half of patients usually recover from dysphagia within a week, testing about 10% of patients with FEES after “more than one day” could represent a bias. What was the higher delay in performing the FEES?
6) In the methods section is not reported the timing of when patients underwent the neurological examination and the endpoints of the clinical examination performed, nor when they performed imaging studies, or who performed them. Please add missing data.
7) I found very interesting the Therapy Requirement Scale. Is it a validated scale for post-stroke dysphagia? If so, please add a reference. If not, this should be its validation study? Please clarify this aspect.
Author Response
To the Editorial Team of Journal of Clinical Medicine, MDPI
Dear Ms. Mance Shi,
We wish to express our appreciation to the Editorial Board and the Reviewers for the careful consideration of our manuscript No. jcm-2022508, entitled ‘A prospective validation study of the functional bedside aspiration screen with endoscopy: Is it clinically applicable in acute stroke?’ and for inviting us to resubmit a version incorporating the revisions suggested by the reviewers. Their valuable feedback helped us to introduce significant improvements and enrich our reference list. Please note the detailed point-by-point response to the reviewers’ comments below.
Enclosed, please find the revised version of the manuscript where all changes have been highlighted in yellow. We thank you again for your time and feedback and we look forward to your final positive response to our revised manuscript.
Sincerely yours,
Dr. Rebecca Kassubek
Corresponding Author
Response to reviewers’ comments
We wish to express our appreciation to the Reviewers for the careful consideration of our manuscript and their insightful comments, which have helped us significantly to improve our manuscript. Enclosed please find the revised version of the manuscript according to the suggestions. Point-by-point responses to the reviewers’ comments are listed below. For readability purposes, we have colored blue the reviewers’ comments.
Reviewer 2:
The prospective validation study by Kassubek and colleagues aims to investigate the reliability of the FBAS 10‐point scale to predict aspiration risk in acute stroke patients in 101 acute ischemic strokes. They observed a 73% sensitivity and 62% specificity 22 for predicting aspiration risk (PAS score ≥6).
Strengths: This study adopted a differentiated FEES protocol which better represents normal mealtime and drinking. In fact, in this study, the sensitivity of the FBAS was lower than published in other clinical assessments before. It is also worth noting that all patients underwent the FEES as the gold standard.
Thus, thanks to the study design, this prospective validation study can paint a very realistic picture of dysphagia and aspiration in acute stroke.
I also appreciated the focus on aspiration, because many bedside screenings are focused on dysphagia without considering aspiration and vice-versa. This is misleading, at best, because dysphagia may occur without aspiration and vice versa.
Authors’ response: Thank you very much for your positive and insightful comments.
Thus, I have just some minor points.
In my opinion, the paper would be considerably improved by reviewing the following minor issues:
Weakness:
Comment 1: 1) I am not sure at all that the observed sensitivity (73%) allows the FBAS to be a “valid first screening instrument to identify patients with high risk for dysphagia” as stated in the text (lines 257-8), even with high specificity.
By definition, a valid swallowing screening in the acute phase of stroke has to be sensitive enough to accurately identify stroke patients at risk of swallowing impairment. A high sensitivity, more than a high specificity, is required because of the increased morbidity and mortality associated with aspiration, which necessitates low false-negative results.
In this study, among 57 patients at low FBAS risk, 13 resulted as false-negative against FEES. This implies a negative predictive value of 77,2%, meaning that almost 1/4 of patients with aspiration will be tested as negative with the FBAS.
- Please, review this statement and report the already validated bedside screenings that have been designed to detect aspiration and achieved a higher sensitivity than FBAS when tested against the FEES (please see Martino R et al. Stroke. 2009;40(2):555-61 and Toscano M et al. Eur J Neurol. 2019 Apr;26(4):596-602).
Authors’ response: We agree with the reviewer´s point. Accordingly, we have removed the term “valid” from the statement, now reading “Therefore, FBAS might serve as a first screening instrument to identify patients with high risk for dysphagia who therefore have a high demand for further intensive, close-meshed surveillance and logopedic therapy.”
In addition, we have included the two studies mentioned in the Introduction (please also refer to comment 5 from Reviewer#1).
Comment 2: 2) I agree with the Author when claiming that one possible explanation for the FBAS's lower sensitivity to detect aspiration is that previous studies probably missed patients who did not show aspiration in FEES with teaspoon volumes.
Anyway, in my opinion, the main reason for the lower sensitivity observed is that the FBAS probably missed the silent aspirators, and likewise most of the bedside dysphagia screenings. Although overt dysphagia is a common feature in stroke patients, those may also suffer from silent aspiration (please see Daniels SK et al. Stroke. 2012;43(3):892-7). This has been linked to reduced laryngopharyngeal sensation, impaired cough reflex, low dopamine or substance P levels, and central/local weakness/incoordination of the pharyngeal musculature (see Toscano M et al. Eur J Neurol. 2019 Apr;26(4):596-602).
- a) - Please, comment on silent aspiration in the discussion as a risk factor (mostly neglected) for PSD and aspiration pneumonia, as well as a possible explanation for the low sensitivity achieved by the FBAS.
- b) -In this regard, since the FEES can detect silent aspiration, it could be of great interest to show data about that or at least, comment on it in the discussion.
Authors’ response: Thank you for highlighting this important point. We have accordingly added important descriptive data in Table 1 and the Results section. Please also note our revised discussion section and our interpretations regarding the lower sensitivity observed with the FBAS in the first paragraphs of the discussion section.
Indeed, all bedside screening tests have an inherited risk for missing silent aspirators. In this regard, they represent limited tools to safely identify all patients with aspiration risk (Ramsey et al., Dysphagia, 2005 Summer;20(3):218-25, Daniels SK et al. Stroke. 2012;43(3):892-7). It has been shown that aspiration without a protective cough increases the incidence of pneumonia to 54% (Nakajoh et al, J Intern Med 2000;247:39–42, Kaneoka et al, Dysphagia, 2018 Apr;33(2):192-199). Screenings incorporating measures of patients’ swallowing reflex and protective cough reflex are important in preventing SAP (Wakasugi et al, Dysphagia 2008, Lee et al, Ann Rehabil Med. 2014 Aug;38(4):476-84).
Our revised text in the discussion in line 331 reads as follows:
“…Besides adequate swallowing reflex, many recent studies highlight that evaluating a patient’s defensive mechanism against airway invasion is important in identifying patients at higher risk for silent aspiration and ultimately, in preventing aspiration pneumonia (ref 44-47). A patient’s possible impairment of the cough reflex and other abnormal protective re-sponses (such as throat clearing) in swallowing trials are incorporated measures in the FBAS protocol.”
In our patient cohort, FEES revealed 15 aspirators with PAS scores 6-8 (14,9%), from whom, a total of 9 patients were asymptomatic during bedside screening (8,9%). Out of those 9 patients, only 3 of them were documented as “low risk” based on the FBAS protocol (FBAS scores 9 or 10). The remaining 6 patients presented with FBAS≤8 (specifically, three patients showed FBAS=8, one showed FBAS=7, one showed FBAS=4 and one FBAS=2).
Maybe in a larger cohort, the aspect of missing silent aspirators would occur more obviously. But as we pointed out, to our opinion using a less detailed protocol without increasing amounts of each consistency, some aspirators might not have been occurred as aspirators, which as a result might have led to a better sensitivity.
Comment 3: 3) In my opinion, when performing a validation study of a bedside screening for aspiration, it is mandatory taking into account the capability of this screening in preventing stroke-associated pneumonia (SAP).
The main reason is that patients with dysphagia may have an 11-fold higher risk of developing pneumonia than non-dysphagic patients, depending on several factors such as stroke severity, the presence of aspiration and silent aspiration, and most of all the kind of bedside screening performed in the acute stage.
In the specific, concerning the screening method, a higher SAP rate is reported: A) in stroke patients who failed a high-sensitive bedside screening compared to those who passed the screening (see Suntrup-Krueger S et al. Cereb. Dis. 2018;45(3-4):101-8). B) but more importantly, in stroke patients who passed a low-sensitive screening for dysphagia compared to those who passed a high-sensitive screening which can also detect both aspiration and silent aspiration. (see Jannini TB et al. Neurol Sci. 2022 Feb;43(2):1167-1176.).
- a) - Please comment on the importance of BSE for preventing SAP in the introduction and discussion
- b) - Please, report the SAP rate in the different groups, otherwise add the lack of this data as a study limitation and highlight that SAPs may arise due, beyond to the aspiration and the kind of BSE, also to stroke-related factors, like immunodepression coupled with tube feeding-related oropharyngeal colonization of pathogenic organisms, especially in the ICUs, where SAP incidence is up to 38% (see Westendorp WF et al. BMC Neurol. 2011;11:110 - Hannawi Y et al. Cerebrovasc Dis 2013;35(5):430-43 - Kishore AK et al. Eur Stroke J. 2019;4(4):318-28).
Authors’ response:
- a) We would like to thank this reviewer for this thoughtful comment. Indeed, there was no case of SAP in our cohort, as now given as a result in the new Table 1. The importance of BSE in lowering or preventing SAP or other adverse dysphagia-related pulmonary sequelae was already mentioned in our introduction however, as suggested, we made sure it was reiterated in our introduction.
The introduction reads,
“It is well established that hospital settings that adhere to formal screening protocols can significantly lower their rates of stroke-associated pneumonia and that conducting an aspiration screen is time-critical.”
Still, the limited sensitivity of FBAS needs to be commented upon as suggested by the reviewer and is now included in the revised discussion:
“Furthermore, this limited sensitivity has to be seen in the context that a bedside screening for aspiration is needed for preventing stroke-associated pneumonia. It is of note that a higher stroke-associated pneumonia rate is reported not only in stroke patients who failed a high-sensitive bedside screening compared to those who passed the screening (Suntrup-Krueger S et al. Cereb. Dis. 2018;45(3-4):101-8), but more importantly, in stroke patients who passed a low-sensitive screening for dysphagia compared to those who passed a high-sensitive screening (Jannini TB et al. Neurol Sci. 2022;43(2):1167-1176.). In our cohort, we did not have any case of stroke-associated pneumonia, possibly attributed to the fact that FEES was performed in each and every patient.”
- b) As already included into our reply to a), there was no case of SAP (0%) in our cohort. This finding can be regarded as the result from FEES being performed in each and every patient.
Comment 4: 4) Why did the Authors exclude hemorrhagic stroke, thus losing possible relevant data? In this regard, it is reported an increased frequency of dysphagia in hemorrhagic stroke patients compared to ischemic ones (Paciaroni M, et al.: Dysphagia following stroke. Eur Neurol 2004;51:162-167). Moreover, hemorrhagic stroke could be also an independent predictor of both post-stroke dysphagia and a persistent pattern of dysphagia at 14 days. Please, comment on this.
Authors’ response: We thank the reviewer for highlighting this important reference and giving us the opportunity to clarify the need for homogeneous sampling.
The accordingly revised Methods section in our manuscript reads,
“Patients with hemorrhagic stroke were not included in this patient cohort. When compared to patients after an ischemic stroke, they are generally more severely affected so that the dysphagia presented after a hemorrhagic stroke is frequently due to additional severe neurological impairments [30-32]. The time window in which swallowing status can be estimated after a hemorrhagic stroke is also highly variable due to frequent compli-cations which may further delay bedside screening. Furthermore, patients with hemor-rhagic stroke are more likely to fail a dysphagia screening - 95% in the study of Joundi et al. (2018) [33-34]. There are no specific risk factors referred to dysphagia which will cause the failure in dysphagia screenings.”
Comment 5: 5) It is reported that FEES was performed with a delay of more than one day in 9.9% of patients (line 136). Since about half of patients usually recover from dysphagia within a week, testing about 10% of patients with FEES after “more than one day” could represent a bias. What was the higher delay in performing the FEES?
Authors’ response: Ninety percent of our patients had FBAS and FEES within 24 hours, unfortunately 10% of patients had not. The greatest delay in performing FEES was four days (the case for four patients). In specific, 72 patients had their endoscopic swallowing evaluation within the first day, 16 patients had their FEES performed on the 2nd day, in nine patients, FEES was conducted in the 3rd day and in four patients on the fourth day post stroke as already stated.
We revised the limitations paragraph accordingly, including the item of transient dysphagia as raised by the reviewer:
“… A further limitation of our study design is based upon the fact that dysphagia in patients with stroke is not only strongly associated with stroke severity but may be even transient in patients with low NIHSS, as demonstrated by Toscano and colleagues (Toscano et al., Eur Neurol. 2015;74(3-4):171-7). For these patients with low NIHSS, frequent dysphagia screenings and re-screenings are mandatory, so that it has to considered that a higher latency between first clinical assessment and FEES can confound the results due to improving dysphagia. As most of our patients had low NIHSS, but 10 percent of our study population had more than 24 hours delay between FBAS and FEES, this latency may be regarded as a confounder to the results in this subgroup.”
Comment 6: 6) In the methods section is not reported the timing of when patients underwent the neurological examination and the endpoints of the clinical examination performed, nor when they performed imaging studies, or who performed them. Please add missing data.
Authors’ response: Accordingly, we have added the following to the methods section,
“All patients have been admitted to the emergency unit of the Department of Neurology, University of Ulm, Ulm, Germany, with the diagnosis of acute stroke. Within standardized operational procedures for stroke admission (Althaus K et al., Ther Adv Neurol Disord. 2021), all patients received the neurological examination (by a board-certified neurologist) and imaging immediately (within 30 minutes) after admission.”
Comment 7: 7) I found very interesting the Therapy Requirement Scale. Is it a validated scale for post-stroke dysphagia? If so, please add a reference. If not, this should be its validation study? Please clarify this aspect.
Authors’ response: Thank you for your comment. The Therapy Requirement Scale has not been validated yet. To our expertise it is a helpful tool to describe and estimate the amount of further therapeutic requirements. It might help to optimize patients` outcomes by quickly organizing not only diet recommendation but other therapeutic options geared to the individual needs of rehabilitation. Our intention was to present this scale for a first time and initiate further validation studies. A new Supplementary Table 1 has been added to clarify this item.

Round 2
Reviewer 1 Report
Authors have replied all my comments ina satisfactory way. I endorse the pubblication of the study.
Reviewer 2 Report
The authors followed all the recommendations given and answered all the doubts raised by modifying the text as required. The manuscript has been greatly improved and, as far as I am concerned, the paper is now suitable for publication.